# Unpaired Visual Editing with Self-Consistent Flow Matching

**Yoad Tewel** [* 1 2]   **Yuval Atzmon** [* 1]   **Gal Chechik** [1]   **Lior Wolf** [2]

## Abstract

Modern generative models possess a deep understanding of visual content, yet training them for image editing typically requires massive datasets of paired examples. This limits scalability, especially for video editing where collecting paired data is prohibitively expensive. We propose a general framework for unpaired training of flow matching editing models. It leverages the base model's knowledge without any external signal. Our approach pairs instruction-following cues extracted from the frozen model with cycle-consistency for structure preservation. To make this tractable, we propose to route gradients from downstream losses over clean predictions to noisy training states. We demonstrate state-of-the-art results on challenging data-scarce image and video editing scenarios. Extensive evaluations and user studies show that our method effectively generalizes to unseen domains and outperforms supervised baselines trained on millions of samples. Analysis reveals that our gradient routing bridges the train-inference gap, and extracting semantic cues from a base model provides a robust training signal that obviates the need for external reward models.

## 1. Introduction

Visual editing is a pivotal task in image and video generation that has been completely transformed in recent years. The current leading editing methods predominantly rely on large datasets of paired examples, requiring explicit source and edited outputs to learn a transformation. Prior work has explored unpaired alternatives: cycle-consistency methods like CycleGAN (Zhu et al., 2017) enable two-domain translation but do not generalize to open-ended instruction-based editing; recent approaches like NP-Edit (Kumari et al., 2025) avoid pairs but rely on external VLM feedback, with

unclear extension to multi-step models or video. Modern generative models already possess the intrinsic capability to generate diverse visual content. This raises a natural question: Is external supervision necessary for learning editing models? This paper explores a different path, leveraging the latent knowledge of a pretrained generative model to learn editing without a single paired example.

Learning to edit without paired data can largely scale the richness and repertoire of editing models. Supervised approaches work well in common editing tasks (Labs et al., 2025; Wu et al., 2025; Bai et al., 2025), but they falter on the long tail of creative edits where ground truth is scarce or impossible to collect. As an example, consider transforming a 2D cartoon into a photorealistic scene, or changing the viscosity of a rushing river to flow like honey - these require before-and-after video pairs that simply do not exist. As visual editing is extended to video, 4D, and beyond, the supervised paradigm becomes increasingly untenable.

Here, we propose a general training framework built on a simple observation: visual editing has two key goals: making the output adhere to the edit instruction, and preserving all aspects of the source except what the instruction explicitly changes. For instruction following, we leverage the frozen base model's knowledge of how edited content should look, as it already understands the dynamics of "cartoon" or "honey-like viscosity". For source preservation, we train the editing model to satisfy cycle consistency (Zhu et al., 2017): editing $\mathbf{x}$ into $\mathbf{y}$, then applying the inverse edit should recover $\mathbf{x}$. Together, these provide complementary training signals from unpaired samples alone.

Realizing this vision for flow-matching models (Lipman et al., 2023), presents fundamental challenges. Standard training corrupts ground-truth outputs with noise to form training inputs. Without paired data, these outputs do not exist, leaving the model with no valid inputs to train on. Furthermore, training operates on noisy intermediate states, while losses like cycle consistency require clean, fully denoised outputs—creating a disconnect that complicates gradient propagation.

We address these challenges through three key components. First, a model unrolling procedure enables the editing model to bootstrap its own training inputs, breaking the chicken-and-egg cycle. Second, we extract instruction-following

---

[*]Equal contribution [1]NVIDIA [2]Tel Aviv University. Correspondence to: Yoad Tewel <yoad.tewel@gmail.com>.

*Proceedings of the 43rd International Conference on Machine Learning*, Seoul, South Korea. PMLR 306, 2026. Copyright 2026 by the author(s).

signal from the frozen base model by isolating the semantic change between source and target prompts—providing supervision without paired data. Third, a gradient-routing mechanism based on Straight-Through Estimation (Bengio et al., 2013) bridges the train-inference gap, allowing training at noisy steps while downstream losses operate on clean outputs. We detail these in Section 4.

We evaluate on both image and video editing, where our method achieves over 75% user preference win rate against supervised baselines trained on millions of pairs. On long-tail style editing, we outperform both supervised and zero-shot methods while generalizing to unseen styles, and remain competitive on general image editing benchmarks. We further provide detailed ablation analysis validating the necessity of each component of our method.

Our contributions are: (1) The first general framework for unpaired training of flow matching editing models. (2) A novel method to query the base model for instruction-following supervision. (3) A novel method to supervise noisy training steps with clean-image losses. It is based on a gradient-routing adaptation of Straight-Through Estimation. (4) State-of-the-art results in unpaired editing, with generalization across video and image domains and user study wins over models trained with large-scale paired data.

## 2. Related Work

**Image and Video Editing:** Obtaining counterfactual training pairs for visual editing is challenging. Most approaches rely on millions of supervised pairs (Labs et al., 2025; Wu et al., 2025; Bai et al., 2025) from synthetic zero-shot methods (Hertz et al., 2023b; Alaluf et al., 2024; Cao et al., 2023; Yang et al., 2025a), simulation (Michel et al., 2023; Yu et al., 2025b), or video-pair extraction (Rotstein et al., 2025; Chen et al., 2025; Song et al., 2023). These risk propagating artifacts, limited to narrow domains, may suffer from realism gaps, or risks having uncontrolled scene changes. In video, zero-shot methods (Geyer et al., 2024; Samuel et al., 2025; Yatim et al., 2025; Lu et al., 2025; Yang et al., 2025b), remain task-specific. Training-based methods require paired supervision, and use synthetic pipelines via key-frame editing (Yu et al., 2025a; Bai et al., 2025), video inpainting (Burgert et al., 2025), and mixed image-video training (Mou et al., 2025; Jiang et al., 2025). Recently, NP-Edit (Kumari et al., 2025) trains without image-editing pairs using an *external* VLM feedback and distribution matching distillation (Yin et al., 2024), with unclear extension to many-step models, or video data that includes motion. We eliminate reliance on external reward models, by leveraging the base model's own knowledge via semantic-guided regularization and cycle consistency, while enabling natural many-step generation.

**Straight-Through Estimation (STE) and Bootstrapped Visual Edits:** STE (Bengio et al., 2013) enables gradient flow through non-differentiable operations in discrete bottlenecks, like Gumbel-Softmax (Jang et al., 2017) and VQ-VAE (van den Oord et al., 2017). In diffusion, DR-Tune (Wu et al., 2024) uses stop-gradients - not STE - so reward propagates via linear sampler updates. Our gradient routing follows the STE pattern and mitigates exposure bias (train-test gap) by providing clean continuous conditioning to the reverse pass while routing gradients through the actual blurry prediction at timestep $t$. We use model rollout to bootstrap edited visuals from the model's own predictions via cycle training, differing from autoregressive self-forcing (Goyal et al., 2016; Huang et al., 2025) which conditions sequentially on past outputs.

**Semantic Guided Directional Regularization:** DDS (Hertz et al., 2023a) is a pixel-space optimization technique that cancels mode-seeking artifacts by contrasting predictions from two distinct image states. In contrast, we query the frozen model on a *single* state with differing prompts to isolate the text-induced velocity shift, focusing pressure where prompts disagree while letting cycle consistency to preserve the common structure. Perp-Neg (Armandpour et al., 2023) employs an unconditional score to extract and subtract only the perpendicular negative component; we subtract source from target queries to obtain a semantic edit-direction, and align via cosine loss.

**Cycle Consistency in Unpaired Translation:** Cycle consistency regularizes unpaired image-to-image translation by enforcing forward-backward reconstruction (Zhu et al., 2017; Liu et al., 2017). CycleNet (Xu et al., 2023) incorporates cycle losses, but swaps input-condition roles in the reverse pass and uses L2 domain regularization; Our reverse pass maintains symmetric roles via gradient routing while our directional regularization enforces semantic alignment with the pretrained model's edit direction. Ouroboros (Sun et al., 2025) relies on paired data and is restricted to single-step generation quality - with unclear extension to multi-step. UNIT-DDPM (Sasaki et al., 2021) jointly trains on two domains with re-noised inputs, and (Wu & De la Torre, 2023; Su et al., 2023; Zhang et al., 2024) rely on zero-shot properties without explicit cycle training.

## 3. Preliminaries - Supervised Image Editing

Flow-matching models (Lipman et al., 2023; Liu et al., 2023) learn to generate data by reversing a noising process. Given a data sample $\mathbf{y}$ and noise $\boldsymbol{\epsilon} \sim \mathcal{N}(\mathbf{0}, \mathbf{I})$, noisy samples are defined as $\mathbf{y}_t = (1 - t)\mathbf{y} + t\boldsymbol{\epsilon}$, where $t=0$ is clean data and $t=1$ is pure noise. A velocity network $\mathbf{G}$ is trained to reverse this process, and at inference, integrates over multiple steps from $t=1$ to $t=0$ to generate samples. In supervised image editing (Figure 1, top), we are given

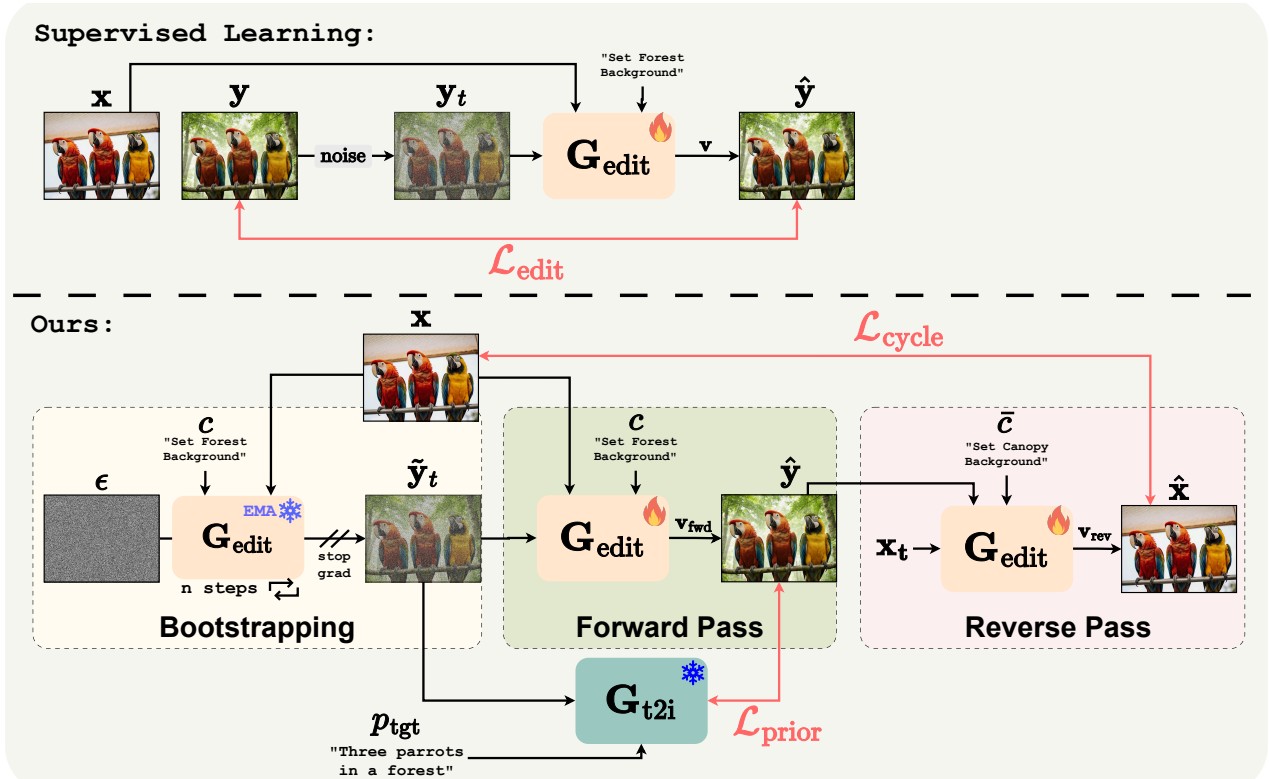

*Figure 1.* **Method overview.** *Top:* Supervised training for image editing. Given a source image $\mathbf{x}$, target image $\mathbf{y}$, and editing instruction $c$, the target is noised to $\mathbf{y}_t$ and fed to the network along with $\mathbf{x}$ and $c$. For clarity, we depict the one-step prediction $\hat{\mathbf{y}}$ supervised against $\mathbf{y}$; the actual loss operates on velocities (Eq. 1). *Bottom:* We finetune a pretrained text-to-image model $\mathbf{G}_{t2i}$ into an editing model $\mathbf{G}_{edit}$, without paired supervision. Given source $\mathbf{x}$ and instruction $c$, a frozen EMA copy of the model generates a noisy pseudo-target $\tilde{\mathbf{y}}_t$ via multi-step sampling. The trainable model then predicts $\hat{\mathbf{y}}$, supervised by: (1) a prior loss aligning the edit direction with $\mathbf{G}_{t2i}$, and (2) a cycle loss reconstructing $\mathbf{x}$ from $\hat{\mathbf{y}}$ using the reverse instruction $\bar{c}$.

a source image $\mathbf{x}$ and a target image $\mathbf{y}$ along with editing instruction $c$. During training, the target image $\mathbf{y}$ is noised to $\mathbf{y}_t$ and fed to the network along with the condition image $\mathbf{x}$ and the editing instruction. The training objective is:

$$\mathcal{L}_{edit} = \mathbb{E}_{t,(\mathbf{x},\mathbf{y}),\epsilon} \left\| \mathbf{u}_t - \mathbf{G}(\mathbf{y}_t, t, c, \mathbf{x}) \right\|^2. \quad (1)$$

where $\mathbf{u}_t = \boldsymbol{\epsilon} - \mathbf{y}$ is the ground-truth velocity. However, obtaining edit pairs at scale is expensive; we develop an unpaired objective that bypasses this limitation.

## 4. Method

We propose a general framework for training flow-matching editing models using only unpaired data and no external supervision (Figure 1). While described for images, the approach applies to any modality; video editing is demonstrated in Section 5. We adapt a pretrained text-to-image model $\mathbf{G}_{t2i}$ into an editing model $\mathbf{G}_{edit}$ conditioned on an edit instruction $c$ and source image $\mathbf{x}$. Training requires only images with source captions ($p_{src}$) and target captions ($p_{tgt}$) describing the original and edited images; no ground-truth edited targets $\mathbf{y}$ are used.

Our framework is built on a simple observation: visual editing has two goals—making the output follow the edit instruction, and preserving all source content except what the instruction changes. In standard supervised training, we achieve this with ground-truth pairs $(\mathbf{x}, \mathbf{y})$: noise the target $\mathbf{y}$ to $\mathbf{y}_t$ and train the model to recover it. In our unpaired setting, we lack $\mathbf{y}$, creating two fundamental gaps: we have no noisy input $\mathbf{y}_t$ to feed the model, and no ground-truth velocity to supervise predictions.

We bridge these gaps by having the model learn from its own predictions, guided by the base model's knowledge. First, we solve the input problem by running the trained model for a few denoising steps to generate noisy edit-targets (Section 4.1), effectively making it bootstrap its own edits. Second, we solve the supervision problem with complementary losses: a *prior loss* leveraging the frozen base model for instruction following (Section 4.2), and a *cycle loss* using the model itself to ensure source preservation (Section 4.3). Finally, for the cycle check to work, the model must see clean images during reconstruction. We enable this via *gradient routing*: verifying edits on high-quality outputs in

the forward pass, while routing gradients to the noisy steps required for training (Section 4.3).

## 4.1. Noisy Input Targets

Flow-matching models take a noisy image as input and predict how to denoise it. In supervised editing, this input is obtained by noising a ground-truth edited image $\mathbf{y}_t$. Without such ground truth, we need an alternative source for the model's input during training.

We generate a pseudo noisy input $\tilde{\mathbf{y}}_t$ using the model itself. Specifically, we maintain a frozen exponential moving average (EMA) copy of the editing model, updated as a moving average of the trainable weights (Grill et al., 2020). At each training step, given source image $\mathbf{x}$, instruction $c$, timestep $t$, and noise $\epsilon$, the EMA model performs $n$ sampling steps from $t=1$ (pure noise) to timestep $t$, producing the noisy input $\tilde{\mathbf{y}}_t$ for the trainable model.

This creates a bootstrapping loop: the model generates noisy inputs, trains on them (supervised by the losses in Sections 4.2 and 4.3), improves, and the EMA copy gradually produces better inputs. The EMA stabilizes this process by smoothing out fluctuations across training steps.

## 4.2. Instruction Following with a T2I base model

Without ground-truth edited images, we need an alternative supervision signal. Our insight: applying an edit instruction to an image should produce a result matching a *target caption* $p_{\text{tgt}}$—a description of what the image would look like after editing. The pretrained T2I model already understands this caption and can guide the edit. Consider the example in Figure 1: we have a source image captioned "three parrots under a canopy" ($p_{\text{src}}$) with instruction "change the background to a forest" ($c$). The target caption is "three parrots in a forest" ($p_{\text{tgt}}$). The T2I model knows how to generate toward this description; we use its velocity as supervision for the editing model.

Formally, given the noisy input $\tilde{\mathbf{y}}_t$, the editing model predicts a denoising velocity $\mathbf{v}_{\text{fwd}} = \mathbf{G}_{\text{edit}}(\tilde{\mathbf{y}}_t, t, c, \mathbf{x})$. To obtain a supervision signal, we query the frozen text-to-image model with the target caption, $\mathbf{v}_{\text{tgt}} = \mathbf{G}_{\text{t2i}}(\tilde{\mathbf{y}}_t, t, p_{\text{tgt}})$. A natural choice would be to directly match this velocity. However, this encourages the model to regenerate the image according to the target caption, often drifting away from the source content and structure.

Instead, we supervise only the *difference* from the source. Let $\mathbf{v}_{\text{src}} = \mathbf{G}_{\text{t2i}}(\tilde{\mathbf{y}}_t, t, p_{\text{src}})$ be the frozen model's velocity for the source caption. We encourage the editing model's velocity to align with the edit direction $\mathbf{v}_{\text{tgt}} - \mathbf{v}_{\text{src}}$, rather than matching $\mathbf{v}_{\text{tgt}}$ absolutely:

$$\mathcal{L}_{\text{dir}} = 1 - \frac{(\mathbf{v}_{\text{fwd}} - \mathbf{v}_{\text{src}}) \cdot (\mathbf{v}_{\text{tgt}} - \mathbf{v}_{\text{src}})}{\|\mathbf{v}_{\text{fwd}} - \mathbf{v}_{\text{src}}\| \, \|\mathbf{v}_{\text{tgt}} - \mathbf{v}_{\text{src}}\|}. \quad (2)$$

This directional loss constrains only the direction of the edit, not its magnitude, which can lead to unbounded velocity norms and training instability. To prevent this, we add a mean squared error term

$$\mathcal{L}_{\text{MSE}} = \|\mathbf{v}_{\text{fwd}} - \mathbf{v}_{\text{tgt}}\|^2$$

that anchors predictions to the frozen model, regulating the magnitude of the edit. The final prior loss is

$$\mathcal{L}_{\text{prior}} = \mathcal{L}_{\text{dir}} + \alpha \mathcal{L}_{\text{MSE}},$$

where $\alpha$ balances directional alignment against velocity magnitude stability.

## 4.3. Source Preservation via Cycle Consistency

The T2I prior encourages instruction-following, but provides no incentive to preserve the source. The directional loss aligns the edit direction with the T2I prior, yet the model could still satisfy it while discarding fine-grained details from the source image $\mathbf{x}$. To enforce source preservation, we employ cycle consistency: a valid edit should be reversible. If we edit $\mathbf{x}$ to produce $\mathbf{y}$, applying the inverse instruction $\bar{c}$ to $\mathbf{y}$ should recover $\mathbf{x}$. While some information loss is inherent to editing, this constraint encourages the model to preserve source content wherever possible: if the forward edit discards information unnecessarily, the reverse pass cannot recover it, increasing the cycle loss.

From the forward velocity $\mathbf{v}_{\text{fwd}}$, we obtain a one-step prediction of the edited image: $\hat{\mathbf{y}} = \tilde{\mathbf{y}}_t - t \cdot \mathbf{v}_{\text{fwd}}$. This prediction then serves as the condition for the reverse pass, which attempts to reconstruct the source. Let $\mathbf{x}_t = (1-t)\mathbf{x} + t\epsilon$ be the noised source. The reverse velocity $\mathbf{v}_{\text{rev}} = \mathbf{G}_{\text{edit}}(\mathbf{x}_t, t, \bar{c}, \hat{\mathbf{y}})$ should recover $\mathbf{x}$—and it can only do so if $\hat{\mathbf{y}}$ retains sufficient information from the source:

$$\mathcal{L}_{\text{cycle}} = \|\mathbf{v}_{\text{rev}} - (\epsilon - \mathbf{x})\|^2. \quad (3)$$

We additionally apply the prior loss symmetrically to the reverse pass, encouraging it to follow the inverse instruction.

**Gradient Routing via Straight-Through Estimation.** A challenge arises because the one-step prediction $\hat{\mathbf{y}}$ is a poor approximation of the result of multi-step denoising, especially for large $t$. Such predictions tend to be blurry and miss fine details (see Figure 6). If the model is conditioned on these degraded estimates during the reverse process at training time, but receives clean images at inference, a train–test mismatch is introduced. In practice, this can cause the model to learn to ignore the conditioning signal.

Thus, we decouple what the model *sees* from what it *learns through*. During the reverse pass, we condition on a clean estimate $\tilde{\mathbf{y}}_0$, obtained by running the EMA model to completion (from $t=1$ to $t=0$). This ensures inputs match inference-time conditions. During backpropagation, gradients bypass $\tilde{\mathbf{y}}_0$ and flow through the one-step prediction $\hat{\mathbf{y}}$:

$$\hat{\mathbf{y}}^{\text{hyb}} = \text{sg}(\tilde{\mathbf{y}}_0) + (\hat{\mathbf{y}} - \text{sg}(\hat{\mathbf{y}})), \qquad (4)$$

where $\text{sg}(\cdot)$ denotes stop-gradient. This allows the forward edit to receive learning signal while keeping conditioning inputs well behaved - adapting Straight-Through Estimation (STE) (Bengio et al., 2013) to the latent denoising setting.

**Identity loss.** The cycle loss supervises reconstruction through a forward-reverse cycle, but assumes the model can already transfer information from the condition. To directly train this capability, we add an identity loss: given the source as both input and condition with the inverse instruction $\bar{c}$—an edit that is already fulfilled—the model should reconstruct it exactly. This teaches faithful preservation of condition information: $\mathcal{L}_{\text{id}} = \|\mathbf{G}_{\text{edit}}(\mathbf{x}_t, t, \bar{c}, \mathbf{x}) - (\boldsymbol{\epsilon} - \mathbf{x})\|^2$.

**Full objective.** The complete loss combines all terms:

$$\mathcal{L} = \mathcal{L}_{\text{cycle}} + \lambda_{\text{prior}}(\mathcal{L}_{\text{prior}}^{\text{fwd}} + \mathcal{L}_{\text{prior}}^{\text{rev}}) + \lambda_{\text{id}}\mathcal{L}_{\text{id}}. \qquad (5)$$

We provide the complete training procedure in Algorithm 1, implementation details in Appendix B, and training data construction details in Appendix D.

# 5. Experiments

Here we evaluate our method on instruction-based image and video editing across long-tail and general-purpose benchmarks, and compare with state-of-the-art methods.

## 5.1. Long-Tail Editing

We evaluate on long-tail scenarios where paired supervision is scarce: unusual image styles and video editing, where collecting aligned pairs is prohibitively expensive.

### 5.1.1. VIDEO EDITING

Our framework naturally extends to video editing by applying it directly to a text-to-video model. We apply our method to the Wan2.2 text-to-video model (Wan et al., 2025). All training objectives—prior, cycle, and identity—remain identical to the image case, applied directly to the video latents. See appendix for additional details.

**Training Data.** We generate unpaired training videos using Wan2.2 (Wan et al., 2025) with captions from VideoUFO (Wang & Yang, 2025), each wrapped in a template specifying a random style (cartoon or photo-realistic).

After filtering videos that fail to match their intended style, we retain 165 cartoon and 163 photo-realistic videos for training (4 each held for validation). Caption preprocessing details are in the appendix.

**Benchmark.** We construct a style-based video editing benchmark from two sources. Real-world videos from Ultra-Video (Xue et al., 2025), we randomly choose videos after labeling by style, yielding 25 cartoon, 25 photo-realistic, and 10 3D-CGI rendered videos, all resized to $480\times832\times81$. The 3D-CGI videos act as out-of-distribution to the training data. We also include 24 cartoon and 25 photo-realistic generated videos, sampled in-distribution. In total: 119 editing tasks across 49 cartoon to photo-realistic, 50 photo-realistic to cartoon, and 20 3D-CGI to {cartoon, photo-realistic}.

**Metrics.** We quantitatively evaluate editing quality through human preference, measuring the average *win rate* of our method against baselines. Participants are asked to judge overall editing quality, considering both application of the target style and preservation of the original content.

**Baselines.** We compare against Ditto (Bai et al., 2025), a recent supervised video editing model trained on one million video editing pairs, by finetuning the WAN-VACE base model (Jiang et al., 2025).

**Evaluation Protocol.** For each input video and instruction, we sample 4 edited videos per method and manually select the best to include in the study. Eight participants completed a user study, each rating 30 comparisons via two-choice questions ("which video shows the better editing result?"), totaling 238 votes. Interface details in Appendix Fig. 10.

**Results.** Despite using no paired data, our method significantly outperforms the supervised baseline (Figure 3), achieving 70.0%±5.4% (SEM) win rate for cartoon targets and 80.5%±2.9% for photo-realistic, averaging 75.3%±2.2% overall. Statistical analysis confirms robustness: a binomial test against chance yields $p < 3\times10^{-15}$; all 8 raters individually preferred our method (per-rater 70–90%); on the 94 majority-vote videos, ours was preferred in 77/94 (sign test, $p < 3\times10^{-10}$); Fleiss' $\kappa = 0.44$ indicates inter-rater agreement well above chance. More striking is generalization: on out-of-distribution 3D-CGI inputs, it wins 85.0% against Ditto's 15.0% – demonstrating the ability of our unpaired approach to transfer to domains never seen during training. Qualitative results in Figure 2 show our method better matches target styles while preserving source content and motion. We provide additional qualitative comparisons in Appendix- Figure 9, and motion videos in the supplemental.

**Quantitative Evaluation.** To complement the user study, we report reference-based video metrics in Table 1: CLIP directional similarity (Brooks et al., 2023) for edit success, DINO (Caron et al., 2021) per-frame similarity for source

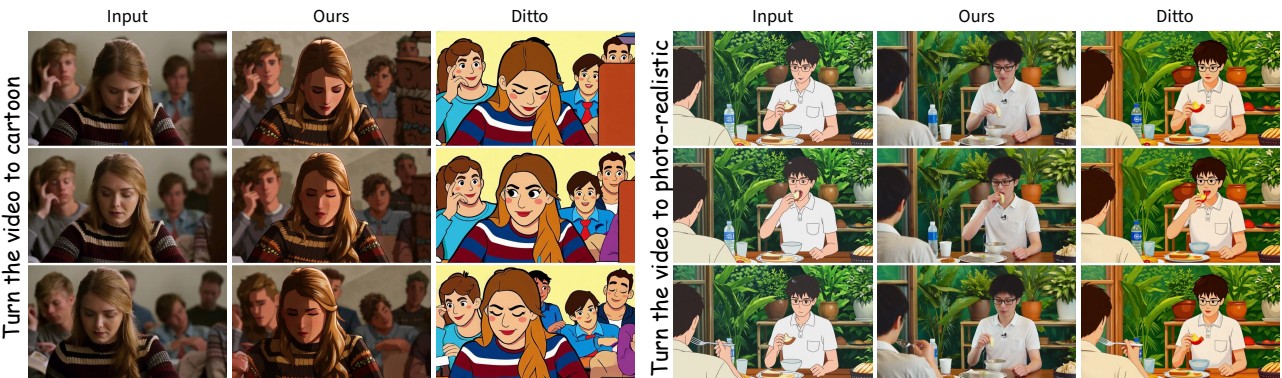

*Figure 2.* Qualitative results on video-editing. Our method better matches the target style while preserving the source content. Several Motion videos are additionally provided in the supplemental material, and also shown in the Appendix - Figure 9.

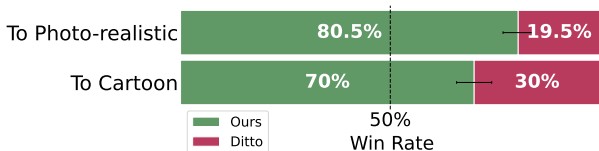

*Figure 3.* User study results on video editing. Users prefer videos generated by our method in both cartoon to photo-realistic editing and photo-realistic to cartoon editing.

preservation, Motion Fidelity (Yatim et al., 2024) for motion preservation, and Dynamic Degree, Aesthetic Quality, and Temporal Flickering from VBench (Huang et al., 2024). Our method outperforms Ditto on edit success, source preservation, and motion fidelity, and matches the source videos on aesthetic quality and temporal flickering. We additionally ablate the contribution of EMA in Appendix C.2.

### 5.1.2. LONG-TAIL STYLE EDITING

**Benchmark.** To evaluate editing types where paired supervision is particularly hard to obtain, we focus on style-based edits: collecting aligned photo↔style pairs is expensive and often infeasible, especially when the target domain has a different visual structure from natural photos (e.g., voxel or low-poly renderings). We construct a long-tail style-editing benchmark with 12 edits across six stylization targets that are not represented in common editing benchmarks: GTA V style, Minecraft style, American comic style, low-poly 3D scene, voxel style, and Lego style. For each style, we evaluate both directions (photorealistic→style and style→photorealistic). For photorealistic→style, we use photorealistic source images from the ImgEdit (Ye et al., 2025) benchmark. For style→photorealistic, we generate stylized source images with Qwen and then edit them back to photorealistic. The benchmark consists of 335 images and 487 edit instructions. Crucially, we do not train our method on any of these styles; our model must generalize.

**Metrics.** Following VIEScore (Ku et al., 2023), we use Qwen2.5-VL-72B (Team, 2024) as an LLM judge to score two aspects on a 1–10 scale: *Semantic Consistency* measures whether the output accurately achieves the requested edit, considering specific visual features; *Perceptual Quality* assesses freedom from artifacts, distortions, or quality degradation. Overall is the geometric mean of both metrics.

**Baselines.** We compare against two strong supervised image editing models trained on millions of paired edits: FLUX-Kontext (Labs et al., 2025) and Qwen-Image-Edit (Wu et al., 2025). We also include FlowEdit (Kulikov et al., 2025) as a zero-shot image editing baseline.

**Results.** Table 2 presents our evaluation. Our method achieves the highest overall score in both directions, with particularly strong gains on the Semantic metric—indicating more accurate style execution. Crucially, we do not train our method on any of these styles, yet it generalizes to them. Qualitative results in Fig. 4 further show that our edits follow the target styles more faithfully than the baselines.

### 5.2. General Image Editing

**Benchmark.** Following prior works, we evaluate on the English subset of GEdit-Bench (Liu et al., 2025), which contains diverse real-world editing instructions (e.g., background, color/material, style, and subject edits).

**Metrics.** We adopt VIEScore (Ku et al., 2023), a reference-free evaluation metric that leverages Qwen2.5-VL-72B (Team, 2024) to assess edit quality. VIEScore evaluates two aspects: Semantic Consistency (SC), which measures alignment between the edit instruction and the output, and Perceptual Quality (PQ), which assesses visual fidelity and artifact-free generation. The Overall score is the geometric mean of SC and PQ; we report this as our primary metric.

*Table 1.* Quantitative video-editing comparison, reported as mean±SEM. *CLIP dir*: CLIP directional similarity (edit success); *DINO Sim.*: per-frame DINO feature similarity to the source (source preservation); *Motion Fidelity*: motion-trajectory consistency with the source; *Dynamic Degree*, *Aesthetic Quality*, and *Temporal Flickering* from VBench.

| Method | CLIP dir ↑ | DINO Sim. ↑ | Motion Fid. ↑ | Dyn. Deg. ↑ | Aesthetic ↑ | Temp. Flicker ↑ |
|---|---|---|---|---|---|---|
| **Ours** | **0.104** ± 0.005 | **0.718** ± 0.012 | **0.715** ± 0.020 | **0.597** ± 0.045 | 0.574 ± 0.007 | 0.967 ± 0.003 |
| Ditto (Bai et al., 2025) | 0.091 ± 0.007 | 0.536 ± 0.017 | 0.616 ± 0.020 | 0.560 ± 0.048 | **0.585** ± 0.007 | **0.972** ± 0.003 |

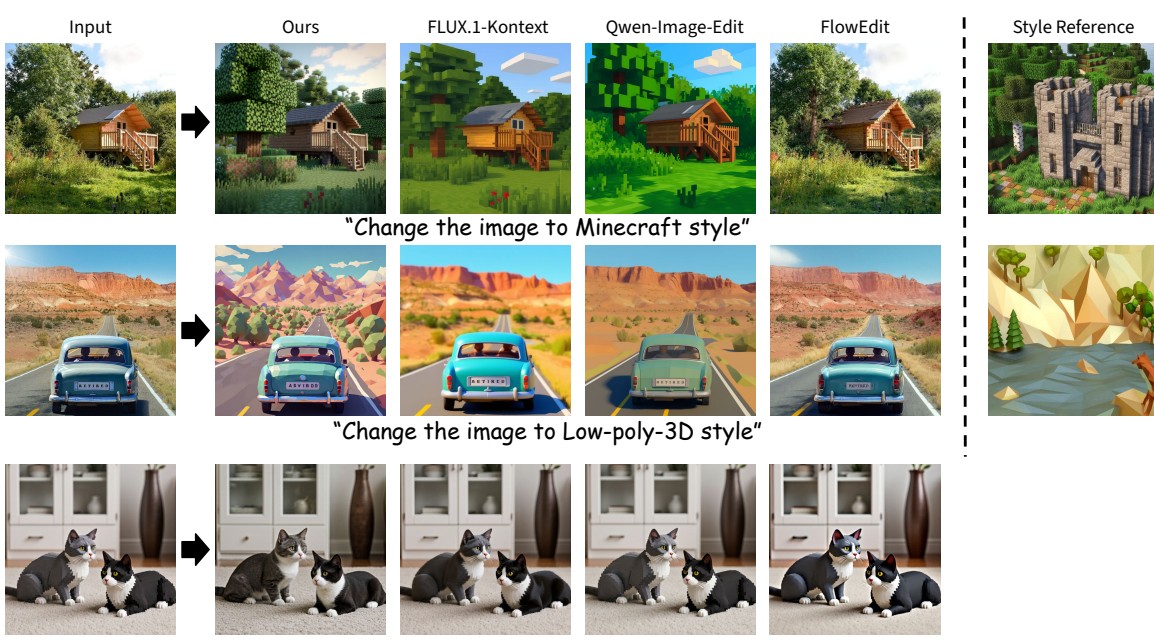

*Figure 4.* Qualitative results on the long-tail style-editing benchmark. Our method better matches the target style while preserving the source content. The "*Style Reference*" column is shown for the reader's convenience and is not used during training or evaluation.

**Baselines.** To isolate the effect of training paradigm, we compare against methods that share our FLUX.1 (Labs, 2024) backbone: FLUX-Kontext (Labs et al., 2025), and FlowEdit (Kulikov et al., 2025).

**Results.** Table 3 presents a breakdown by edit category. Our unpaired method is competitive with FLUX-Kontext across most categories, notably outperforming it on motion changes, human-centric edits, and style changes: categories where paired supervision may be limited. Kontext excels at subject removal and text editing, where precise paired examples provide a clear advantage. Qualitative results (Fig. 7 - 8 in supplementary) show that our edits are often more realistic than Kontext, likely due to synthetic artifacts in supervised paired training data, for example, turning a statue into jade yields a more plausible material appearance. To complement VIEScore with a fine-grained measure of structural fidelity, in Appendix C.1 we additionally report SSIM/PSNR/LPIPS on *unedited* pixels.

## 6. Ablation Study

We ablate key components of our method on GEdit-Bench, using VIEScore (Ku et al., 2023) with Qwen2.5-VL-72B (Team, 2024) as the evaluator. We report quantitative results: *Edit Success* (whether the output reflects the requested edit) and *Source Preservation* (whether unedited regions are preserved) in Table 4, and provide qualitative comparisons in Fig. 5.

**Source Preservation.** As shown in Table 4 and Fig. 5, removing the cycle loss reduces source preservation, consistent with the role of the cycle constraint in encouraging the forward edit to retain information required for reverse reconstruction. Removing gradient routing similarly degrades preservation due to a train–test mismatch (Fig. 6): during training, the reverse pass is conditioned on noisy one-step predictions, which weakens the conditioning signal and can cause the model to rely on it less. Finally, removing the directional term (retaining only the MSE component of the prior) increases drift from the source by more strongly

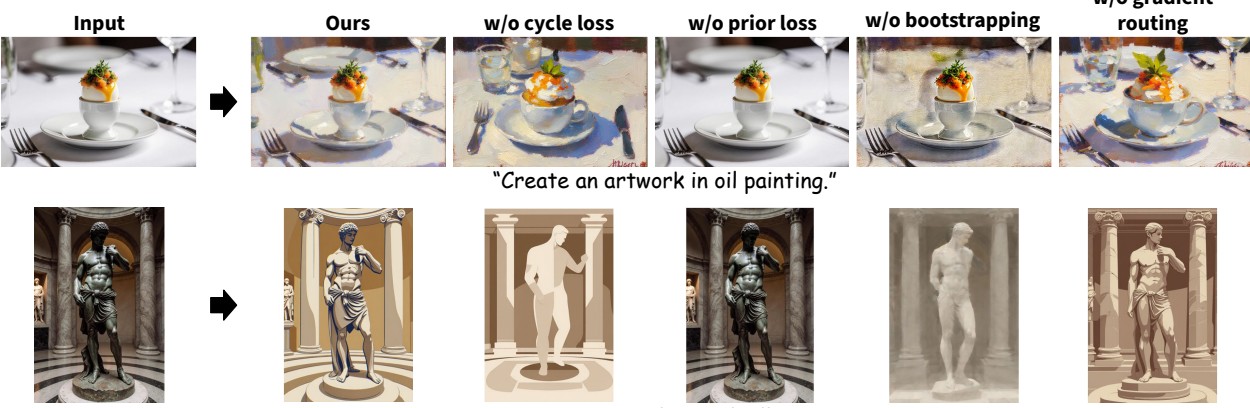

*Figure 5.* Qualitative ablation results. Removing gradient routing, cycle loss, or directional loss leads to stronger edits but degrades source preservation (visible in fine details and background consistency). Without bootstrapping, edits become unreliable. Without regularization, the model collapsed to identity mapping, preserving the source unchanged.

*Table 2.* Style transfer evaluation on six long-tail styles (GTA V, Minecraft, American comic, low-poly 3D, voxel, Lego). Our method, trained without style-specific paired data, outperforms supervised and zero-shot baselines in both directions.

| Method | Semantic ↑ | Quality ↑ | Overall ↑ |
|---|---|---|---|
| *Style → Photorealistic* | | | |
| **Ours** | **7.67** | **8.99** | **8.30** |
| Kontext | 6.87 | 8.96 | 7.85 |
| Qwen-Image-Edit | 6.86 | 8.76 | 7.75 |
| FlowEdit | 4.27 | 8.90 | 6.20 |
| *Photorealistic → Style* | | | |
| **Ours** | **5.22** | 7.67 | **6.33** |
| Kontext | 3.97 | 6.73 | 6.00 |
| Qwen-Image-Edit | 4.87 | 7.39 | 5.17 |
| FlowEdit | 1.46 | **9.09** | 3.64 |

*Table 3.* Performance breakdown by edit category on GEdit-Bench (Overall score, higher is better). Best results in **bold**.

| Edit Type | Ours | Kontext | FlowEdit |
|---|---|---|---|
| Background change | 6.69 | **6.93** | 2.17 |
| Color alter | **7.55** | 7.47 | 3.54 |
| Material alter | **5.54** | 5.50 | 1.77 |
| Motion change | **6.88** | 4.53 | 5.24 |
| Human edit | **6.59** | 3.94 | 5.40 |
| Style change | **6.95** | 5.90 | 2.86 |
| Subject add | **7.36** | 6.88 | 3.99 |
| Subject remove | 1.91 | **6.94** | 2.43 |
| Subject replace | 5.74 | **6.17** | 4.06 |
| Text change | 2.10 | **5.44** | 2.84 |
| Tone transfer | 5.98 | **6.10** | 5.20 |

pulling the output toward the target instruction.

**Instruction Following.** Removing the regularization loss leads to an identity-collapse failure mode: the model attains high source preservation while producing minimal change, resulting in very low edit success (Table 4), which is also evident qualitatively (Fig. 5).

**Training Stability.** Bootstrapping provides stable training inputs that match the forward model's expected distribution (a noisy version of the *edited* image). Without bootstrapping, the forward process is instead driven by a noised source image (e.g., $\mathbf{x}_t$). It introduces a distribution mismatch and destabilizes training, degrading both edit success and source preservation (Table 4) and producing edit artifacts (Fig. 5). Random identity steps act as a regularizer, not a stability requirement: removing them leaves edit success high and only modestly degrades source preservation (Table 4).

**Prior Loss Decomposition and $\alpha$ Sensitivity.** The prior loss $\mathcal{L}_{\text{prior}} = \mathcal{L}_{\text{dir}} + \alpha\,\mathcal{L}_{\text{MSE}}$ couples a directional alignment term with an MSE anchor that regulates velocity magnitude. To isolate their roles, we evaluate each term separately and sweep $\alpha$ in Table 5. $\mathcal{L}_{\text{dir}}$ is the primary driver of source preservation (7.701 vs. 7.233 when using $\mathcal{L}_{\text{MSE}}$ alone), while $\mathcal{L}_{\text{MSE}}$ is the primary driver of edit success (8.400 vs. 8.117 when using $\mathcal{L}_{\text{dir}}$ alone); the two terms are thus complementary rather than redundant. Sweeping intermediate values ($\alpha \in \{0.02, 0.05, 0.1\}$) traces a smooth trade-off curve: larger $\alpha$ improves edit success but degrades source preservation.

## 7. Limitations

Our method inherits the knowledge and biases of the pre-trained base model. If the base model lacks understanding of a target domain, our method cannot reliably edit toward it. A trade-off of our caption-based supervision is weaker

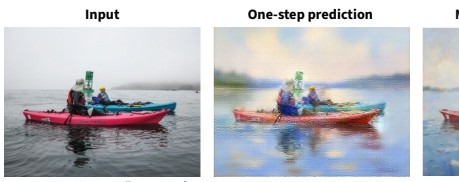

| Input | One-step prediction | Multi-step prediction |

"Turn the image into an impressionist painting"

*Figure 6.* Comparison of one-step vs. multi-step predictions. One-step predictions tend to be blurry and lack fine details, while multi-step sampling produces clean outputs. Our gradient routing conditions on the clean multi-step estimate while backpropagating through the one-step prediction.

*Table 4.* Ablation study on GEdit-Bench. We evaluate the contribution of each component to edit success and source preservation. Removing gradient routing, cycle loss, or directional loss improves edit success at the cost of source preservation. Without bootstrapping, both metrics degrade. Without regularization loss, the model collapses to identity. Random identity steps are not required to prevent collapse but improve source preservation.

| Method | Edit Success ↑ | Source Pres. ↑ |
|---|---|---|
| **Ours (full)** | 8.317 | 7.617 |
| w/o gradient routing | 8.917 | 7.183 |
| w/o cycle loss | 8.983 | 7.233 |
| w/o directional loss | 8.400 | 7.233 |
| w/o bootstrapping | 5.517 | 7.050 |
| w/o regularization loss | 0.633 | 9.767 |
| w/o random identity steps | 8.413 | 7.450 |

performance on object removal (Table 3). This stems from how we derive supervision from the T2I prior: the target caption $p_{tgt}$ describes the scene *after* removal, but simply omits the object rather than explicitly describing its absence. For example, removing a cat from "a cat on a sofa" yields "a sofa", the caption provides no explicit signal that the cat should be removed, only that it is no longer mentioned. This weaker supervisory signal makes removal edits harder to learn compared to additive or transformative edits.

## 8. Conclusion

We presented a framework for training visual editing models without paired supervision. By combining instruction-following cues from a frozen text-to-image model with cycle-consistency constraints, our approach learns to edit images and videos using only unpaired data. A key technical contribution is gradient routing, which bridges the train-inference gap by conditioning on clean predictions while backpropagating through noisy states. Experiments demonstrate that our unpaired method matches or outperforms supervised baselines trained on millions of paired examples, while generalizing to unseen domains.

*Table 5.* Decomposition of the prior loss $\mathcal{L}_{prior} = \mathcal{L}_{dir} + \alpha \mathcal{L}_{MSE}$ on GEdit-Bench. $\mathcal{L}_{dir}$ is the primary driver of source preservation, while $\mathcal{L}_{MSE}$ is the primary driver of edit success. Varying $\alpha$ traces a smooth trade-off curve without inducing instability.

| Prior variant | Edit Success ↑ | Source Pres. ↑ |
|---|---|---|
| $\mathcal{L}_{MSE}$ only | 8.400 | 7.233 |
| $\mathcal{L}_{dir}$ only ($\alpha=0$) | 8.117 | 7.701 |
| Mixed ($\alpha=0.02$, **Ours**) | 8.317 | 7.617 |
| Mixed ($\alpha=0.05$) | 8.403 | 7.468 |
| Mixed ($\alpha=0.1$) | 8.566 | 7.300 |

## Acknowledgements

We thank Assaf Shocher, Lior Hirsch and Omri Kaduri for helpful discussions.

## Impact Statement

This paper presents work whose goal is to advance the field of Machine Learning. As with other generative and editing technologies, our method could potentially be misused to create misleading visual content. We encourage the development of detection methods and responsible use guidelines alongside editing capabilities. We do not foresee other societal consequences that must be specifically highlighted here.

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

## A. Training Algorithm

Algorithm 1 provides the complete training procedure for our method.

---

**Algorithm 1** Self-Consistent Flow Editing Training

---

**input** Dataset $\mathcal{X} = \{(\mathbf{x}, p_{\text{src}}, p_{\text{tgt}}, c, \bar{c})\}$, text-to-image model $\mathbf{G}_{\text{t2i}}$
**output** Editing model $\mathbf{G}_{\text{edit}}$

1: **while** training **do**
2:     $(\mathbf{x}, p_{\text{src}}, p_{\text{tgt}}, c, \bar{c}) \sim \mathcal{X}, \quad t \sim \mathcal{U}(0, 1), \quad \epsilon \sim \mathcal{N}(\mathbf{0}, \mathbf{I})$
3:
4:     *// Generate pseudo-targets via EMA model (Section 4.1)*
5:     $\tilde{\mathbf{y}}_1 \leftarrow \epsilon$
6:     **for** $s = 1$ to $0$ with step $\Delta s = 1/n$ **do**
7:         $\tilde{\mathbf{y}}_{s-\Delta s} \leftarrow \tilde{\mathbf{y}}_s - \Delta s \cdot \mathbf{G}_{\text{EMA}}(\tilde{\mathbf{y}}_s, s, c, \mathbf{x})$
8:         **if** $s = t$ **then**
9:             $\tilde{\mathbf{y}}_t \leftarrow \tilde{\mathbf{y}}_s$
10:         **end if**
11:     **end for**
12:
13:     *// Forward pass: predict edit velocity*
14:     $\mathbf{v}_{\text{fwd}} \leftarrow \mathbf{G}_{\text{edit}}(\tilde{\mathbf{y}}_t, t, c, \mathbf{x})$
15:     $\hat{\mathbf{y}} \leftarrow \tilde{\mathbf{y}}_t - t \cdot \mathbf{v}_{\text{fwd}}$                                      *// One-step prediction*
16:
17:     *// Prior loss: align with T2I edit direction (Section 4.2)*
18:     $\mathbf{v}_{\text{src}} \leftarrow \mathbf{G}_{\text{t2i}}(\tilde{\mathbf{y}}_t, t, p_{\text{src}}), \quad \mathbf{v}_{\text{tgt}} \leftarrow \mathbf{G}_{\text{t2i}}(\tilde{\mathbf{y}}_t, t, p_{\text{tgt}})$
19:     $\mathcal{L}_{\text{dir}}^{\text{fwd}} \leftarrow 1 - \text{cosine\_sim}(\mathbf{v}_{\text{fwd}} - \mathbf{v}_{\text{src}}, \mathbf{v}_{\text{tgt}} - \mathbf{v}_{\text{src}})$
20:     $\mathcal{L}_{\text{MSE}}^{\text{fwd}} \leftarrow \|\mathbf{v}_{\text{fwd}} - \mathbf{v}_{\text{tgt}}\|^2$
21:     $\mathcal{L}_{\text{prior}}^{\text{fwd}} \leftarrow \mathcal{L}_{\text{dir}}^{\text{fwd}} + \alpha \mathcal{L}_{\text{MSE}}^{\text{fwd}}$
22:
23:     *// Cycle loss: reconstruct source from edit (Section 4.3)*
24:     $\mathbf{x}_t \leftarrow (1 - t)\mathbf{x} + t\epsilon$
25:     $\hat{\mathbf{y}}^{\text{hyb}} \leftarrow \text{sg}(\tilde{\mathbf{y}}_0) + (\hat{\mathbf{y}} - \text{sg}(\hat{\mathbf{y}}))$                          *// Gradient routing*
26:     $\mathbf{v}_{\text{rev}} \leftarrow \mathbf{G}_{\text{edit}}(\mathbf{x}_t, t, \bar{c}, \hat{\mathbf{y}}^{\text{hyb}})$
27:     $\mathcal{L}_{\text{cycle}} \leftarrow \|\mathbf{v}_{\text{rev}} - (\epsilon - \mathbf{x})\|^2$
28:     Compute $\mathcal{L}_{\text{prior}}^{\text{rev}}$ analogously to $\mathcal{L}_{\text{prior}}^{\text{fwd}}$ for reverse pass
29:
30:     *// Identity loss: preserve condition information*
31:     $\mathcal{L}_{\text{id}} \leftarrow \|\mathbf{G}_{\text{edit}}(\mathbf{x}_t, t, \bar{c}, \mathbf{x}) - (\epsilon - \mathbf{x})\|^2$
32:
33:     *// Total loss and update*
34:     $\mathcal{L} \leftarrow \mathcal{L}_{\text{cycle}} + \lambda_{\text{prior}}(\mathcal{L}_{\text{prior}}^{\text{fwd}} + \mathcal{L}_{\text{prior}}^{\text{rev}}) + \lambda_{\text{id}}\mathcal{L}_{\text{id}}$
35:
36:     Update $\mathbf{G}_{\text{edit}}$ with $\nabla\mathcal{L}$; update $\mathbf{G}_{\text{EMA}}$ as moving average of $\mathbf{G}_{\text{edit}}$
37: **end while**

---

## B. Implementation Details

### B.1. Image Editing

**Architecture.** We construct an image editing model $\mathbf{G}_{\text{edit}}$ by finetuning a pretrained FLUX.1-dev (Labs, 2024) text-to-image model. FLUX.1-dev is desgined to generate images from text prompts. To support conditioning on the source image $\mathbf{x}$, we concatenate the VAE encoding of the source image to the noisy target latent along the token sequence dimension. We finetune the model using LoRA (Low-Rank Adaptation) (Hu et al., 2022) with rank 64.

**Training.**   We follow the training procedure in Algorithm 1 with a learning rate of $3e^{-4}$, batch size 8, and 30000 training steps. We optimize with AdamW (Loshchilov & Hutter, 2019) using weight decay $10^{-2}$. We set $\lambda_{\text{prior}} = 1.0$, $\lambda_{\text{id}} = 0.2$, and $\lambda_{\text{cycle}} = 1.0$. For stability, we train for the first 200 steps using only the identity loss, ensuring the model can propagate information from the source image to the output. Throughout training, we additionally sample identity steps with probability 15%; in these steps we replace the instruction with the inverse instruction and train the model to reconstruct the source image. We use 10 integration steps for bootstrapping the noisy target during training.

**Computation.**   We train the model on 8 H100 GPUs with a batch size of 1 per GPU. Our method incurs additional overhead relative to supervised training, taking $\sim 3\times$ longer per step (2.9s vs. 0.97s). However, we find that training converges in substantially fewer steps: the model exhibits meaningful editing capabilities after as few as 1000 training steps. During inference, our method incurs no additional overhead compared to other editing models, and we sample with 20 inference steps.

## B.2. Video Editing

**Architecture Adaptation.**   To adapt the Wan2.2 text-to-video model (Wan et al., 2025) for editing, we modify its input conditioning mechanism. The Wan2.2 architecture processes video latents as a sequence of tokens. To condition the model on the source video $\mathbf{x}$, we concatenate the source latents with the noisy input latents $\mathbf{y}_t$ along the token dimension (channel dimension concatenation is also possible depending on specific implementation, but token concatenation is standard for Transformers).

Critically, we assign the source video tokens the *same positional encodings* as the noisy input tokens. This forces the model to treat the condition and input as spatially and temporally aligned counterparts. The model learns to distinguish between the two based on their noise levels: the source video $\mathbf{x}$ is always clean (noise-free), while the input $\mathbf{y}_t$ is noisy according to timestep $t$. This allows the attention mechanism to attend to the corresponding clean features in the source video when denoising the target.

All other aspects of the architecture, including the temporal attention layers, remain unchanged. The training objectives (prior, cycle, identity) are applied to the video latents exactly as they are for images.

**Video Training Setup.**   We fine-tune the Wan2.2 text-to-video model using LoRA with rank 64. We ablate EMA in Appendix C.2. Training was performed on 8 H100 GPUs with a batch size of 8 (1 per GPU) for 750 steps (approximately 1 minute per step), using a learning rate of $10^{-4}$ with videos scaled to 320x576 resolution. We applied an identity loss probability (id-chance) of 10%, where the instruction is replaced with the inverse instruction and the model is trained to reconstruct the source video. Additionally, we sample timesteps using logit-normal sampling with shift (Esser et al., 2024), with shift parameter $s = 12$ to bias toward higher noise levels. Our training pipeline is adapted from Musubi Tuner (Kohya-ss, 2025).

**Inference.**   For inference, we use the UniPC sampler (Zhao et al., 2023) with the default configuration provided by Wan2.2 at native 480x832 resolution.

## C. Additional Results

### C.1. Structure Preservation on Unedited Regions

VIEScore captures *semantic* agreement with the instruction and overall perceptual quality, but it does not directly measure fine-grained spatial fidelity of regions that the instruction does not touch. To complement the VIEScore numbers in the main paper, we evaluate background structure preservation with low-level reference-based metrics computed exclusively on *unedited* pixels.

**Protocol.**   We use the localizable edit categories of GEdit-Bench (Liu et al., 2025) (color_alter, material_alter, subject_remove, subject_replace, text_change), for which the instruction targets a well-defined object or region. Since GEdit-Bench does not provide ground-truth edit masks, we obtain them automatically with Lang-SAM (Medeiros, 2023) applied to the instruction's target noun (e.g., "car" for "change the color of the car"). For each input image, the *same* mask is applied to the outputs of every method, so all methods are scored against an identical definition of the unedited region. We then compute SSIM, PSNR, and LPIPS on the complement of the mask, between the source image and each edited output.

| **Input** | **FLUX.1-Kontext** | **FlowEdit** | **Ours** |
|---|---|---|---|

"Adjust the image style to a watercolor effect"

"Replace with jade"

"Adjust the background to a garden"

*Figure 7.* Additional qualitative comparisons on the general image editing benchmark (GEdit-Bench). We compare our method against FLUX-Kontext and FlowEdit. Our results are often more realistic than Kontext (which can inherit artifacts from synthetic paired training data) while following the instruction more faithfully than the zero-shot baseline.

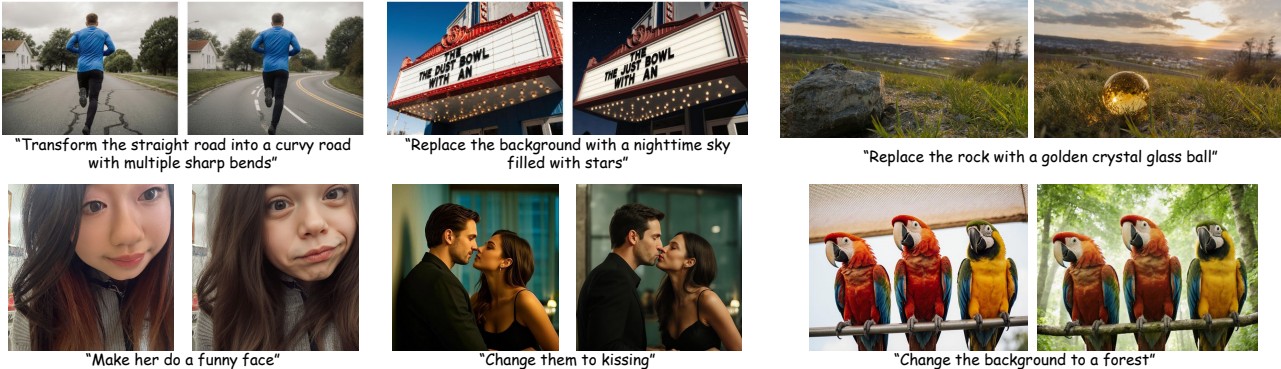

"Transform the straight road into a curvy road with multiple sharp bends"

"Replace the background with a nighttime sky filled with stars"

"Replace the rock with a golden crystal glass ball"

"Make her do a funny face"

"Change them to kissing"

"Change the background to a forest"

*Figure 8.* Additional qualitative results of our method on the general image editing benchmark (GEdit-Bench).

**Results.** Table 6 reports the three metrics on the unedited regions. Our method significantly outperforms the zero-shot FlowEdit baseline on all three metrics. Despite being trained *without any paired data*, our structure preservation is broadly comparable to FLUX-Kontext, which is trained on millions of supervised editing pairs and therefore benefits from explicit pixel-level alignment between source and target. These results indicate that the cycle-consistency objective, together with gradient routing on clean predictions, provides a competitive structural prior even in the absence of pixel-aligned supervision.

*Table 6.* Structure preservation on unedited regions of GEdit-Bench (localizable edits).

| Method | SSIM ↑ | PSNR ↑ | LPIPS ↓ |
|---|---|---|---|
| **Ours** | 0.861 | 24.09 | 0.103 |
| FlowEdit (Kulikov et al., 2025) | 0.703 | 19.54 | 0.180 |
| FLUX-Kontext (Labs et al., 2025) | 0.921 | 27.20 | 0.077 |

### C.2. EMA Ablation in Video Editing

Our default video configuration omits EMA, and the qualitative results and user study in the main paper use this non-EMA variant. Here we evaluate an EMA variant: pseudo-targets are generated using an exponential moving average of the training weights rather than a stop-gradient copy of the current weights. Enabling EMA improves source preservation (DINO Sim.) and motion fidelity at a small cost in editability (CLIP dir).

*Table 7.* EMA ablation on video editing, reported as mean±SEM. Enabling EMA improves source preservation and motion fidelity at a small cost in editability.

| Method | CLIP dir ↑ | DINO Sim. ↑ | Motion Fid. ↑ | Dyn. Deg. ↑ | Aesthetic ↑ | Temp. Flicker ↑ |
|---|---|---|---|---|---|---|
| **Ours w/ EMA** | $0.104 \pm 0.005$ | $\mathbf{0.718} \pm 0.012$ | $\mathbf{0.715} \pm 0.020$ | $\mathbf{0.597} \pm 0.045$ | $0.574 \pm 0.007$ | $0.967 \pm 0.003$ |
| Ours w/o EMA | $\mathbf{0.110} \pm 0.005$ | $0.656 \pm 0.011$ | $0.687 \pm 0.022$ | $0.571 \pm 0.046$ | $0.550 \pm 0.007$ | $0.969 \pm 0.003$ |
| Ditto (Bai et al., 2025) | $0.091 \pm 0.007$ | $0.536 \pm 0.017$ | $0.616 \pm 0.020$ | $0.560 \pm 0.048$ | $\mathbf{0.585} \pm 0.007$ | $\mathbf{0.972} \pm 0.003$ |

## D. Data Construction

This section describes how we construct the *training* tuples used by our unpaired objective.

**Training tuple format.** For each training example, we construct a tuple $(\mathbf{x}, p_{\mathrm{src}}, p_{\mathrm{tgt}}, c, \bar{c})$ consisting of a source image $\mathbf{x}$, a source prompt $p_{\mathrm{src}}$, a target prompt $p_{\mathrm{tgt}}$, a forward edit instruction $c$, and a reverse instruction $\bar{c}$. Intuitively, $p_{\mathrm{tgt}}$ describes the desired edited image, while $\bar{c}$ specifies the inverse transformation to recover the source.

### D.1. Image Data

**Data generation with a VLM.** Given a large collection of captioned images, we generate edit specifications using a vision-language model (VLM), applied independently to each image. For an input image $\mathbf{x}$, the VLM outputs a JSON object with five fields: `edit_type`, `src_caption`, `instruction`, `tgt_caption`, and `reverse_instruction`. We then set $p_{\mathrm{src}} \leftarrow$ `src_caption`, $c \leftarrow$ `instruction`, $p_{\mathrm{tgt}} \leftarrow$ `tgt_caption`, and $\bar{c} \leftarrow$ `reverse_instruction`. Concretely, we use Qwen3-VL-30B-A3B-Thinking, and run our generation pipeline on a subset of the OpenImages dataset (Krasin et al., 2017).

**Prompting details and constraints.** To diversify supervision, we define an edit taxonomy and require the VLM to select *exactly one* edit type per image from: color change, texture/material change, shape adjustment, add, remove, replace, background change, style change, action/pose change, and text manipulation. Each category description emphasizes centrality (affect a prominent object or the entire background) and distinctness (the change should be visually noticeable). For style edits, we provide a fixed list of artistic styles (e.g., watercolor, oil painting, anime, ukiyo-e, charcoal). For each image, we sample a small subset of categories (typically $k{=}3$) and present them as the only allowed choices, requesting a single JSON output. To balance the dataset, category sampling is weighted by (i) a fixed priority that slightly over-samples categories such as `remove` and `replace`, and (ii) an online usage-balancing term that favors categories used less often so far in the current run.

We include explicit prompt rules to ensure compatibility with our objective. If metadata indicates a non-photorealistic image, we occasionally force a style-conversion edit with probability $p_{\text{style}}=0.15$, restricting the edit type to `style_change` and setting the target style to photorealistic. The reverse instruction $\bar{c}$ must be self-contained and cannot reference an unknown "original" state (we forbid phrases such as "restore", "revert", or "undo"); instead it must directly specify the inverse transformation to apply to the edited image (e.g., "change the car color to red"). The target caption $p_{\text{tgt}}$ must describe only the final edited image, without temporal language (e.g., "now", "changed to") and without negative phrasing (we discourage "without X"). The source caption $p_{\text{src}}$ must describe the image as observed, and if the image is stylized (non-photorealistic), the style is explicitly included.

Finally, we parse the VLM output as JSON (robustly handling markdown code blocks) and keep an example only if all required fields are present.

### D.2. Video Data

**Caption Preprocessing Pipeline.**    We preprocess captions for video generation in three steps:

1. Randomly sample detailed captions from the VideoUFO dataset (Wang & Yang, 2025).

2. Use Qwen3-VL-30B-A3B-Instruct in text-only mode to sanitize captions by removing any existing style hints.

3. Wrap sanitized captions in a template that appends a randomly selected style (cartoon or photo-realistic).

Generated videos are manually verified to match the intended style; mismatches are discarded.

**Benchmark Construction.**    For real-world videos, we randomly sample from UltraVideo (Xue et al., 2025) after assigning style labels (cartoon, photo-realistic, or 3D-CGI) through manual inspection of a larger candidate pool. Final benchmark videos are randomly drawn from each style category. All videos are resized to $480 \times 832$ resolution with 81 frames.

## E. User Study Details

We conducted a user study to evaluate video editing quality, collecting 238 total votes from 8 participants. Each participant was presented with a reference video, an editing instruction (e.g., "Turn the video to photo-realistic style"), and two edited videos (Options A and B) produced by different methods. Participants were asked: "Which video is the better editing result?" and instructed that a good edit should: (1) follow the target style specified in the instruction, and (2) preserve the content and structure of the original video. The assignment of methods to Options A and B was randomized. A screenshot of the interface is shown in Fig. 10.

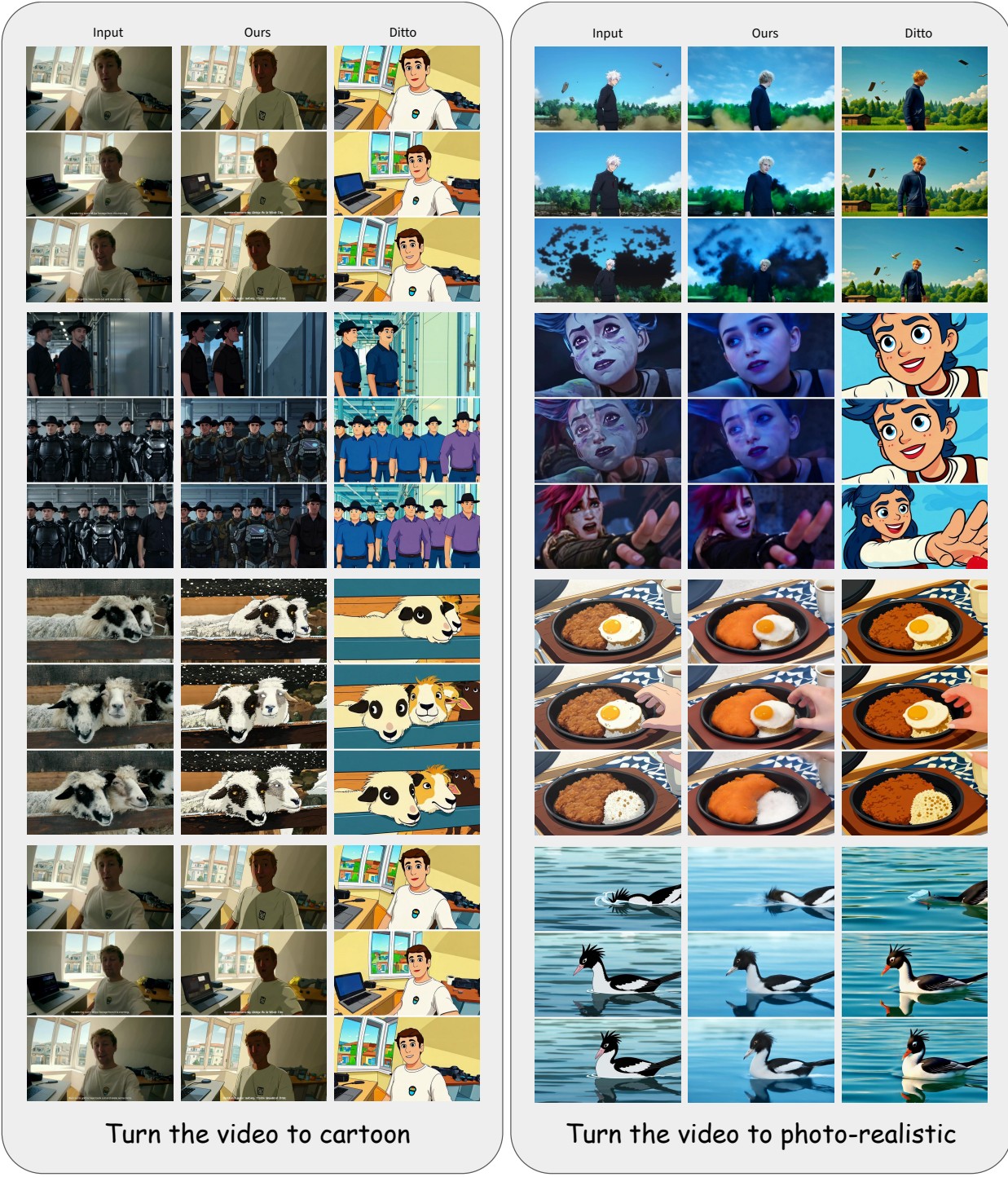

*Figure 9.* Additional qualitative comparisons on video-editing. Our method better matches the target style while preserving content.

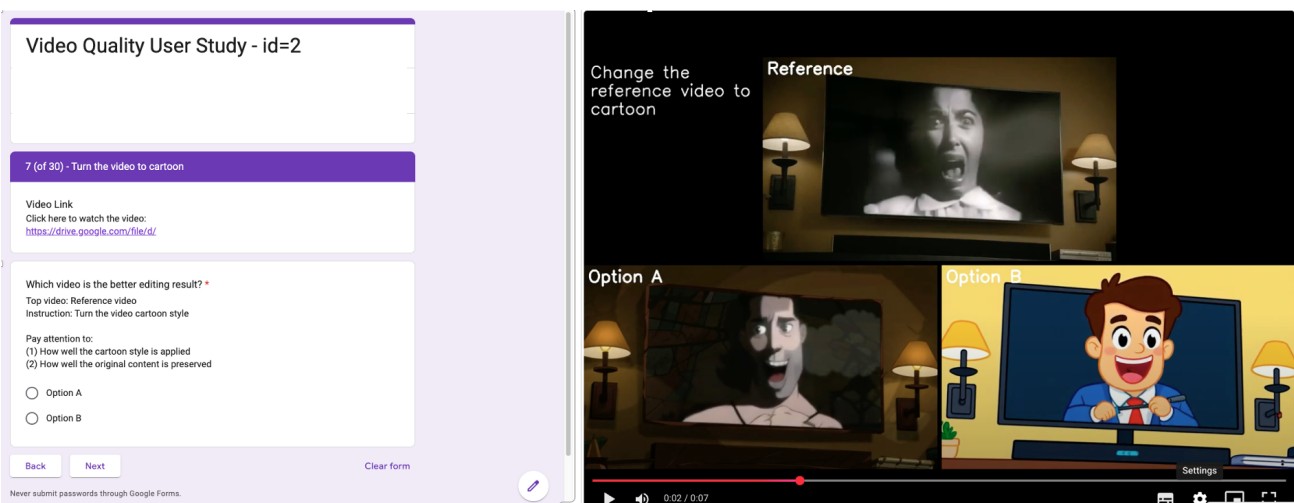

*Figure 10.* A screenshot of the user-study interface

