# OpenReview forum: "Unpaired Visual Editing with Self-Consistent Flow Matching"
_ICML.cc/2026/Conference — ICML 2026 regular_

### Official Review · Reviewer_ks1A · 2026-03-12

**Soundness:** 3
**Presentation:** 2
**Significance:** 2
**Originality:** 2
**Overall Recommendation:** 4
**Confidence:** 3

**Summary:**

The paper proposes an unpaired training framework for flow-matching image and video editing that removes the need for paired (source, target) edits. It combines two complementary signals: (i) a semantic, instruction-following prior extracted from a frozen text-to-image (T2I) base model via a directional velocity loss (target-minus-source prompt difference) and (ii) source-preserving cycle consistency enforced through a forward–reverse editing loop. A key technical ingredient is a gradient-routing scheme inspired by Straight-Through Estimation that conditions the reverse pass on clean predictions while backpropagating through noisy one-step estimates, thereby reducing the train–inference gap. The method achieves strong results on long-tail image style edits and video editing.

**Compliance With Llm Reviewing Policy:**

Affirmed.

**Final Justification:**

The rebuttal addresses my main concerns and improves the overall soundness of the paper. Although the presentation still requires improvement, my final recommendation is to accept the submission.

**Key Questions For Authors:**

1. How sensitive is performance to the α weight in the prior loss (directional + MSE)? Can you provide a sweep and report on edit success vs. source preservation trade-offs?
2. Did you attempt comparisons with NP-Edit or related unpaired instruction editors? If not feasible, can you discuss expected differences and whether your method could be combined with reward-based signals?
3. How does the method perform on localized edits (small regions) versus global edits? Any failure cases with strong locality where cycle consistency may be less informative?

**Limitations:**

Yes.

**Strengths And Weaknesses:**

## Strength
1. The problem framing and decomposition of objectives are clear; the method flow and training algorithms are well explained with equations and a step-by-step algorithm.
2. The framework is general and could be applied to different modalities and backbones, which has the potential to shift practice away from heavily synthetic or supervised pipelines.

## Weakness
1. My main concern is that the reliance on a VLM to produce training tuples (source/target captions, forward/reverse instructions) weakens the claim of “no external supervision" and introduces dependence on the VLM’s quality and biases.
2. Appendix B.2 states you did not use EMA. How then are pseudo-targets ỹt in Algorithm 1 and Section 4.1 generated for the forward pass? Please clarify the training inputs and report an ablation on EMA vs. non-EMA.
3. The directional prior still includes an MSE-to-target term that may encourage drift; sensitivity to α and stability trade-offs are not deeply explored. Moreover, the term “regularization loss” in ablations appears to refer to the prior loss, but it is not consistently named; clearer terminology would avoid confusion.
4. The experiment scope needs to be extended. The human study for video is small (8 raters, 238 votes) and selects the best of 4 samples per method, which introduces potential selection bias; statistical rigor (e.g., inter-rater agreement, per-video significance) is limited.

---

> ### Author Rebuttal · Authors · 2026-03-31
>
> We are encouraged that the reviewer acknowledged the "**practice shift**" and "**generality**" of the framework, "**key**" technical ingredients, and "**strong results**". We value your feedback and will update the manuscript accordingly.
>
> ## (1a) VLM weakens "no external supervision"
> To clarify, by "no external supervision" we mean that a VLM does not provide rewards or gradients. Additionally - no visual pairs (source, edited-target) are required.
>
> ## (1b) Training tuples depend on VLM quality and bias
> We acknowledge VLM biases may affect text tuples, but we rely on it *only for text*; visual learning comes from two self-derived signals: a prior loss and a cycle reconstruction loss. In contrast, supervised or VLM-guided methods rely on both visual and textual signals, injecting much stronger visual biases. For example, in Figure 7, FLUX-Kontext produces an unrealistic green for "Replace with jade". In our case, the visual signal is grounded in the base model's manifold, since the VLM only provides text descriptions.
>
> Furthermore, the VLM is replaceable. Our framework can operate using existing captioned image datasets or structured captioning methods (Describe Anything, Lian et al.) paired with an LLM or rule-based templates to generate the edit instructions.
>
> ## (2) Video did not use EMA, clarify, report an ablation
> EMA improves performance but is not essential. In Appendix B.2, pseudo-targets are generated using n denoising steps with a stop-gradient copy of the current training weights rather than a moving average of past weights. As video results were already strong without EMA we did not ablate over it. We now provide the requested ablation; errors are S.E.M.
>
> We report edit success (CLIP dir [Brooks et al., 2023]), per-frame visual similarity (DINO Sim. [Caron et al., 2021]), Motion Fidelity [Yatim et al., 2024], and relevant VBench metrics [Huang et al., 2023].
>
> | Method | CLIP dir ↑ | DINO Sim. ↑ | Motion Fidelity ↑ | Dynamic Degree ↑ | Aesthetic Quality ↑ | Temporal Flickering ↑ |
> |---|---|---|---|---|---|---|
> | Ours w/ EMA | 0.104±0.005 | **0.718**±0.012 | **0.715**±0.020 | **0.597**±0.045 | 0.574±0.007 | 0.967±0.003 |
> | Ours w/o EMA | **0.110**±0.005 | 0.656±0.011 | 0.687±0.022 | 0.571±0.046 | 0.550±0.007 | 0.969±0.003 |
> | Ditto | 0.091±0.007 | 0.536±0.017 | 0.616±0.020 | 0.560±0.048 | **0.585**±0.007 | **0.972**±0.003 |
>
> EMA improves source preservation (DINO Sim., Motion Fidelity) with a slight trade-off in editability (CLIP dir.). Both variants outperform Ditto across most metrics and match source videos in Aesthetic Quality and Temporal Flickering (0.569 and 0.965).
>
> ## (3) Extend experiment scope
> In (2) above, we provide a new quantitative comparison to Ditto across six metrics on video editing. All metrics are computed on a single random generation per prompt.
>
> ## (4) Human study: potential selection bias and statistical rigor
> Additional statistical analysis confirms robustness: binomial test for preference above chance yields p < 3×10⁻¹⁵; all 8 raters prefer our method (70%–90%). Raters agreed on method in 79% of videos (Fleiss' κ = 0.44, 0 = chance [Landis & Koch, 1977]). Among 94 videos with majority, ours was preferred in 77/94 (sign test, p < 3×10⁻¹⁰). Best-of-4 selection was applied symmetrically to both methods; additionally, the new analysis in (2) uses a single seed per video, eliminating selection bias.
>
> ## (5) Sensitivity to α in L_prior
> See response (1) to Reviewer F1R4.
>
> ## (6) Expected differences to NP-Edit, combining with reward-based signals
> NP-Edit is a recent work with unreleased code, preventing direct comparison; our evaluation includes FlowEdit as a related unpaired baseline.
>
> **Multi-step and domain generality.** NP-Edit relies on external VLM rewards, which require clean images to produce reliable scores. This restricts NP-Edit to few-step distilled models, making it not applicable to SOTA multi-step generators. It is also difficult to extend to domains like video, where reliable reward models are scarce. Our gradient routing enables supervision at any noise level without external rewards.
>
> **Expected differences.** We expect our generative prior to produce more realistic outputs, as edits are grounded in the base model's learned manifold. VLM rewards are expected to better guide complex edits that require reasoning (e.g. causality or physics).
>
> **Combining with reward signals.** Our gradient routing could enable external rewards to operate on clean predictions while training multi-step models.
>
> ## (7) Method performance on localized edits vs. global edits
> Our method performs well on both. Table 2 in paper reports results per edit category, including global edits (e.g. style) and localized edits (e.g. color alter). We consistently outperform Kontext on global edits and match on localized categories. Masked SSIM/PSNR/LPIPS over unedited regions (see response (3) to F1R4) further confirm background preservation. We did not observe failure cases with strong locality.

---

> > ### Author Rebuttal · Reviewer_ks1A · 2026-04-01
> >
> > Thank the authors for providing the additional experiments and explanations. After reading the response, I have one minor suggestion. Please improve the presentation in the revised main paper to ensure the discussed issues are completely clear; **incorporating additional visualizations for the motivation and the framework would be helpful**. Since my major concerns have been successfully addressed, this minor point does not affect my overall assessment. Therefore, I will raise my score to 4.

---

> > > ### Author Response · Authors · 2026-04-07
> > >
> > > Dear Reviewer ks1A,
> > >
> > > We thank you for your time, the constructive feedback provided throughout the review process, and for confirming that your major concerns have been successfully addressed.
> > >
> > > In the revised manuscript, we will refine the text to ensure the issues discussed in the rebuttal are completely clear. We will also incorporate additional visualizations to better convey the motivation and the proposed framework.
> > >
> > > Overall, we’re pleased to see unanimous acceptance scores from all reviewers. Reviewers highlighted the importance of the problem and identified our framework as a strong technical contribution with the potential to shift practice in unpaired editing.

---

### Official Review · Reviewer_tr1M · 2026-03-12

**Soundness:** 3
**Presentation:** 3
**Significance:** 1
**Originality:** 3
**Overall Recommendation:** 4
**Confidence:** 3

**Summary:**

This paper proposes a framework for training flow-matching models for image and video editing without paired data by combining semantic guidance from a frozen text-to-image model with a cycle-consistency objective. To stabilize training and bridge the train-inference gap, the method creatively introduces a bootstrapping loop for noisy inputs and a gradient-routing mechanism based on Straight-Through Estimation.

**Compliance With Llm Reviewing Policy:**

Affirmed.

**Final Justification:**

(a) Fully resolved - My concerns have been adequately addressed.

**Key Questions For Authors:**

-  In the video editing adaptation (Appendix B.2), does the architecture rely entirely on the base Wan2.2 model's temporal attention to maintain consistency, or were any explicit temporal consistency regularizers tested? Could the lack of an explicit temporal loss explain the observed flickering in stylized outputs?
- How crucial are the randomly sampled "identity steps" (15% probability for images, 10% for video) to preventing model collapse? Did you ablate training entirely without them after the initial warmup steps to see if the cycle loss alone is sufficient to retain source information?

**Limitations:**

Requires a T2I model for the target domain

**Strengths And Weaknesses:**

# Strengths

- The authors appear to examine a pressing problem in generative AI: the scalability bottleneck caused by the reliance on massive paired datasets for image and particularly video editing. Moving toward unpaired training unlocks potential for complex, long-tail tasks.
- The authors proceed to present the central challenge of applying cycle consistency to flow-matching models and offer a straightforward yet effective solution via gradient routing and cycle loss to enforce source structure preservation.

# Weaknesses

- The biggest weakness is the assumption that paired data is unavailable while the framework simultaneously relies heavily on a robust pretrained T2I model. If a strong T2I model exists for the target domain, it is highly feasible to generate vast amounts of synthetic paired data using intermediate spatial condition signals (e.g., Canny edges, depth maps, or CLIP image features via ControlNet-like architectures). This alternative paired-data generation pipeline is simple, accessible, and widely used, which calls into question whether the proposed complex cycle-consistency training is strictly necessary to achieve these results.
- The visual quality of the generated videos leaves room for improvement. Specifically, the generated results (such as the cartoon style transfer) exhibit strong flickering and temporal instability, suggesting the current approach struggles to maintain temporal consistency without explicit motion supervision.

---

> ### Author Rebuttal · Authors · 2026-03-31
>
> We are encouraged that the reviewer acknowledged our technical contributions to be **creative** and **effective**, and our setup to be **a pressing problem**. We value your feedback and will update the manuscript accordingly.
>
> ## (1) Is the unpaired data paradigm necessary given accessible synthetic pair pipelines?
>
> Our results show that synthetic pair pipelines have clear limitations in domains where paired data is harder to obtain.
>
> On long-tail style editing (Table 1), our unpaired method outperforms Kontext and Qwen-Image-Edit, both trained on millions of synthetic pairs, across all six unseen styles. This suggests that synthetic pair generation struggles to cover the full diversity of visual domains. The gap is even more pronounced in video editing (Figure 3), where our method achieves a 75% user preference win rate over Ditto, a supervised model trained on one million video editing pairs. Collecting or synthesizing aligned video pairs with consistent motion, temporal coherence, and faithful style transfer is fundamentally harder than the image case, and current pipelines (key-frame editing, video inpainting) introduce artifacts that propagate into training. For example, in Figure 7 FLUX-Kontext renders an unrealistic color for "Replace with jade".
>
> That said, the reviewer raises a valid point that synthetic pipelines are effective for common editing tasks, and we view these paradigms as complementary rather than competing. Our unpaired framework could serve as a data generation engine for supervised methods, producing diverse pseudo-pairs in domains where synthetic pipelines fall short. Conversely, supervised signals could be mixed with our cycle and prior losses to combine the strengths of both approaches.
>
> ## (2) Visual quality of generated videos leaves room for improvement
> To address these concerns, we provide a new quantitative evaluation (see response (2) to Reviewer ks1A). By comparing both the EMA and non-EMA versions, we see that using EMA in video editing further improves visual quality and temporal consistency. We note the qualitative results in the submission used the non-EMA variant.
>
> Specifically, the EMA variant achieves aesthetic and temporal flickering scores comparable to the original source videos and the Ditto baseline. Additionally, it outperforms Ditto in edit success (CLIP dir: 0.104 vs 0.091), source preservation (DINO Sim: 0.718 vs 0.536), and motion alignment (Motion Fidelity: 0.715 vs 0.616). We achieve these results without relying on paired video datasets or external reward models. While visual quality has room to improve, we note our framework represents an early step in the unpaired editing paradigm.
>
> ## (3) Were any explicit temporal consistency regularizers tested?
>
> Our current formulation only used Wan2.2's native temporal attention to enforce consistency across frames. To keep the method general and uniform across both image and video domains, we deliberately did not experiment with video-specific temporal losses or complex inference-time attention sharing constraints. While this design choice preserves the framework's simplicity, future work can further improve temporal stability by integrating explicit temporal regularizers.
>
> ## (4) How crucial are randomly sampled identity steps to preventing mode collapse?
>
> Following the reviewer's suggestion, we ablated the random identity steps on GEdit-Bench to isolate their contribution and test if the model collapses without them.
>
> | Method | Edit Success ↑ | Source Preservation ↑ |
> |---|---|---|
> | **Ours**, full method | 8.317 | 7.617 |
> | **Ours**, w/o random identity steps | 8.413 | 7.450 |
>
> As shown, removing the random identity steps does not lead to model collapse. This demonstrates that the cycle loss and directional prior loss alone are indeed sufficient to retain source information. Omitting these steps does degrade Source Preservation. Identity steps are therefore not strictly required for preventing collapse but act as an effective regularizer to better anchor unedited regions. We will include this ablation in the revised paper.

---

> > ### Author Rebuttal · Reviewer_tr1M · 2026-04-01
> >
> > n/a

---

> > > ### Author Response · Authors · 2026-04-07
> > >
> > > Dear Reviewer tr1M,
> > >
> > > We thank you for your time, the constructive feedback provided throughout the review process, and for acknowledging that our rebuttal fully resolved your concerns.
> > >
> > > We’re pleased to see unanimous acceptance scores from all reviewers. Reviewers highlighted the importance of the problem and identified our gradient routing mechanism as a strong technical contribution with the potential to shift practice in unpaired editing.
> > >
> > > We’re glad the rebuttal addressed your concerns, and that there’s a positive consensus on the paper’s impact. We hope this will be taken into account in your final score.

---

### Official Review · Reviewer_F1R4 · 2026-03-13

**Soundness:** 3
**Presentation:** 3
**Significance:** 3
**Originality:** 3
**Overall Recommendation:** 4
**Confidence:** 5

**Summary:**

This paper proposes a general framework for training flow-matching visual editing models without paired data. It leverages a frozen text-to-image base model to extract semantic editing directions (using the velocity difference between target and source prompts) and employs cycle-consistency to preserve unedited structures. To resolve the conflict between the clean outputs needed for the cycle-consistency check and the noisy states required for training, the authors introduce a clever gradient routing mechanism based on Straight-Through Estimation. Additionally, the model bootstraps its own noisy training targets via an EMA model. The method achieves state-of-the-art performance in data-scarce video and long-tail image editing scenarios.

**Compliance With Llm Reviewing Policy:**

Affirmed.

**Final Justification:**

The authors have satisfactorily addressed my first two concerns, and I find the core unpaired methodology highly inspiring. However, I remain skeptical about the claim that its structure preservation is comparable to FLUX-Kontext, as the paper's visual cases show noticeable background and subject inconsistencies in non-stylization tasks. I will maintain a positive score, but I expect the authors to include explicit visual comparisons and candidly discuss these consistency limitations in the final manuscript.

**Key Questions For Authors:**

¸● Q1: Could you provide a more granular ablation study on $L_{prior}$? Specifically, how does the model perform when trained only with $L_{MSE}$ versus only with $L_{dir}$? Furthermore, how sensitive is the training stability and edit quality to the balancing parameter $\alpha$?
● Q2: Given the heavy computational burden of the EMA multi-step rollout (especially for video generation), have you explored ways to optimize this process? For instance, how does reducing the number of integration steps ($n$) for the EMA model affect both the training speed and the final edit quality?
● Q3: To better reflect the unintended pixel shifts and background inconsistencies observed in the qualitative results (e.g., Figure 8) , would you consider supplementing the VIEScore with metrics specifically designed to evaluate structure/background preservation? Metrics such as PSNR, SSIM, or LPIPS calculated explicitly over the unedited masks would provide a more rigorous quantitative assessment of source consistency.

**Limitations:**

yes

**Strengths And Weaknesses:**

Strengths:
● The paper addresses a highly critical bottleneck in generative visual editing: the prohibitive cost of collecting paired training data, especially for video and long-tail style edits.
● The proposed methodology is elegant and technically sound. Specifically, the use of a velocity difference for semantic guidance successfully enforces instruction-following without destroying the source structure. Furthermore, the gradient routing mechanism is a brilliant technical contribution that effectively bridges the train-inference gap inherent in latent denoising.
● Empirically, the paper delivers exceptionally strong results; it not only outperforms heavily supervised models trained on millions of pairs (e.g., Ditto, FLUX-Kontext) but also demonstrates remarkable zero-shot generalization to unseen out-of-distribution domains, such as 3D-CGI videos. Finally, the comprehensive ablation study rigorously validates the necessity of each framework component, making the authors' claims highly credible

Weaknesses:
● W1: Incomplete analysis of the prior loss ($L_{prior}$) formulation. While the directional loss ($L_{dir}$) is conceptually elegant for aligning semantic shifts, its distinct empirical contribution relative to the MSE constraint ($L_{MSE}$) remains somewhat ambiguous. Although Table 3 ablates the directional loss, the paper lacks a comprehensive decomposition (e.g., evaluating $L_{MSE}$ in isolation vs. $L_{dir}$ in isolation) and an analysis of the hyperparameter $\alpha$. It is crucial to understand whether $L_{dir}$ is the primary driver of semantic alignment or if $L_{MSE}$ dominates the loss landscape.
● W2: Severe computational overhead during training. The proposed "Gradient Routing" and bootstrapping mechanism relies on multi-step EMA sampling to generate clean targets at every training iteration. While the authors transparently report the hardware and time costs in Appendix B (e.g., a 3x slowdown for images and ~1 minute per step for videos on 8 H100 GPUs), this substantial bottleneck severely limits the scalability and accessibility of the framework. The paper currently lacks a discussion on potential acceleration strategies or the trade-off between the number of EMA integration steps and final model quality.
● W3: Noticeable degradation in source consistency and unintended pixel shifts. Although the unpaired paradigm is highly novel, the qualitative results reveal obvious artifacts that compromise visual fidelity. As observed in Figure 8, there are unintended color shifts on subjects, structural pixel drift (e.g., col 3), and background inconsistencies (e.g., row 2, col 2) . These fine-grained spatial and identity preservation failures are not adequately captured by the primary VIEScore metric, indicating a disconnect between the quantitative evaluation and the actual perceptual quality of the non-edited regions.

---

> ### Author Rebuttal · Authors · 2026-03-31
>
> We appreciate the kind words - the reviewer found our technical contributions to be **brilliant**, our empirical results to be **exceptionally strong** and **remarkable**, and our problem setup to be **highly critical**. We value your feedback and will update the manuscript accordingly.
>
> ## (1) Does L\_dir or L\_MSE dominate the loss landscape? Add a more complete analysis.
>
> Neither loss dominates - they serve complementary roles.
> Following your feedback, we conducted a comprehensive analysis of the L\_prior formulation on GEdit-Bench to isolate the contributions of L\_dir and L\_MSE, as well as to sweep the balancing parameter α for L\_prior = L\_dir + α · L\_MSE.
>
> As shown below, L\_dir is the primary driver of source preservation (7.701 vs 7.233 without it), while L\_MSE is the primary driver of edit success (8.400 vs 8.117 without it).
>
> | Method (L\_prior) | Edit Success ↑ | Source Preservation ↑ |
> |---|---|---|
> | L\_MSE only | 8.400 | 7.233 |
> | L\_dir only (α=0) | 8.117 | 7.701 |
> | Mixed (α=0.02, **Ours**) | 8.317 | 7.617 |
> | Mixed (α=0.05) | 8.403 | 7.468 |
> | Mixed (α=0.1) | 8.566 | 7.300 |
>
> The α hyper-parameter balances the two losses and as a result, provides a trade-off of the two metrics. We find that training remains stable across different values of α; rather than causing sudden failures, varying α predictably spans the trade-off curve. Higher α values lead to better editability, but can hurt Source Preservation.
>
> ## (2) Computational overhead during training and analysis of the number of EMA steps
>
> Our method incurs per-step overhead relative to supervised training - but as shown in the Appendix, converges in substantially fewer steps:
> the model exhibits meaningful editing capabilities after as few as 1000 training steps. For comparison, Ditto trains for ~16,000 steps on 64 GPUs with 1M paired videos, on top of VACE which itself required additional supervised fine-tuning. This may be partly due to the intrinsic training signal being more consistent than noisy synthetic pairs, allowing faster convergence.
>
> In addition, we conducted an analysis over the EMA integration steps (n) that produce the noisy input ỹ\_t:
>
> | Method | Δ Edit Success | Δ Source Pres. |
> |---|---|---|
> | n=1 | -0.100 | +0.134 |
> | n=2 | -0.017 | +0.034 |
> | n=3 | +0.050 | -0.066 |
> | n=5 (**Ours**) | 0.000 (8.317) | 0.000 (7.617) |
> | n=10 | +0.100 | -0.100 |
>
> Decreasing the number of integration steps linearly accelerates training but weakens instruction-following. Conversely, increasing the steps (e.g., n=10) improves edit success but slightly degrades background preservation. We selected n=5 to balance editability and structural preservation while significantly reducing the computational bottleneck.
>
> Finally, regarding potential acceleration strategies. Our framework is completely orthogonal to recent advancements in fast sampling. Integrating a few-step distilled base model would reduce the EMA rollout cost. We will add this discussion to the revised paper.
>
> ## (3) Evaluate structure preservation to quantify unintended pixel shifts and source inconsistencies
>
> Our method's source consistency is comparable to FLUX-Kontext and significantly outperforms FlowEdit, despite being trained without paired data. To rigorously assess this, we evaluated samples from GEdit-Bench with localizable edits (color/material alter, subject remove/replace, and text change). Since GEdit-Bench lacks ground-truth masks, we used Lang-SAM (GroundingDINO + SAM 2.1) to segment the instruction's target object (e.g., segmenting "car" for "change the color of the car"). We then computed SSIM, PSNR, and LPIPS on the unedited pixels (the complement of the mask), applying identical masks across all methods.
>
> Note that our method, which is trained without paired data, is compared with methods that were trained using millions of exact supervised pairs for precise localized editing.
>
> | Method | SSIM ↑ | PSNR ↑ | LPIPS ↓ |
> |---|---|---|---|
> | Flux Kontext | 0.921 | 27.20 | 0.077 |
> | **Ours** | 0.861 | 24.09 | 0.103 |
> | FlowEdit | 0.703 | 19.54 | 0.180 |
>
> We hope this unpaired approach will encourage the field to develop better models that do not rely on prohibitively expensive paired datasets. We will include this evaluation in the revised paper.

---

### Decision · Program_Chairs · 2026-04-30

**Decision:**

Accept (regular)

**Comment:**

This paper proposed a framework to leverage unpaired data to train editing model. It designed a pair of cycled editing tasks to train the model with a cycle-consistency loss. A T2I model is used to support a prior loss served as regularization in the training process.

After the rebuttal period, all three reviewers recommended towards acceptance. The authors did a good job addressing reviewers' concerns and clarified the technical details.

We congratulate the authors on the acceptance of their paper and encourage them to revise the paper to incorporate the comments from the rebuttal period in the final version.